# Mountain Arable Land Abandonment (1968–2018) in the Romanian Carpathians: Environmental Conflicts and Sustainability Issues

**Ionuț Săvulescu, Bogdan-Andrei Mihai, Marina Vîrghileanu * , Constantin Nistor and Bogdan Olariu**

Faculty of Geography, University of Bucharest, 010041 București, Romania; savulescu@geo.unibuc.ro (I.S.); bogdanandrei0771@gmail.com (B.-A.M.); constantin@geo.unibuc.ro (C.N.); bogdanolariu28@gmail.com (B.O.)
* Correspondence: marina.virghileanu@geo.unibuc.ro

**Abstract:** The agricultural mountain landscape in the Romanian Carpathians follows the same change trend in other European mountains, from variety and individuality to simplification and uniformization. Our paper proposes two complementary case studies from the Southern Carpathians—Poiana Mărului and Fundata, representative areas for the entire Carpathian ecoregion. The research focuses on a remote sensing approach with Corona KH-4B (1968) and Planet Scope (2018) images at 2.0–3.0 m resolution used for mapping arable plots pattern and size change. Landscape transformation modelling is focused on four-hectare sampled grid for both case study areas, followed by a landscape metric analysis. Fundata area is the most transformed, where arable plots disappeared under the service-based economy pressure. Poiana Mărului shows an earlier stage of landscape transformation, where the arable land abandonment process is incipient. The spatial and statistical analysis and field survey confirmed that tourism changed the traditional agricultural landscape, generating potential environmental conflicts and indicating the sustainability degree.

**Keywords:** traditional agriculture; transformation; tourism; sustainability; environmental conflicts; CORONA KH-4B; Planet Scope

## 1. Introduction

Traditional agriculture plays a leading role for biodiversity conservation [1] and food security [2,3] and helps the preservation of the local identity of the regions [4]. Traditional agriculture abandonment is an important process that occurs worldwide, primarily affecting less productive, remote, and mountainous areas [5,6]. The literature offers various definitions for agricultural land abandonment process; however, the most common definition refers to a patch that was previously used for crops, pasture, orchards, vineyards, etc. but that no longer preserves that function and has not been converted into another land cover class [7]. The intensification of traditional sustainable agriculture abandonment [8] increases the pressure upon biodiversity and ecosystems, generating environmental conflicts [9,10].

Statistical data shows that more than 11% of the European Union territory (about 20 million hectares) are under high potential risk of abandonment by 2030, with the most affected areas located in Southern and Eastern Romania, Southwestern France, Southern and central Spain, Portugal, Cyprus, Poland, Latvia, and Estonia [11]. The agricultural land use change in European mountain regions involves local and external actors [12], with common interests up to a limit from which environmental conflicts occur and new scenarios are to be considered [13]. Different regional studies explained the specific problems: the conflict between traditional mountain farming and the mass tourism as in Austrian Alps [14],

the conflict between local communities and tourism investors in protected areas or natural parks [15], the conflict between development and traditional mountain life with social and environmental consequences as in Spanish Pyrinees [16] or in Swiss Alps [17], or the loss of agricultural land in mountain areas under the pressure of tourism [18].

The risk of agricultural land abandonment in the European Union [11] is calculated on the basis of aggregated statistical indicators like the land market, the farm income, the investment volume in farming, the age of farm holder and the population density [19]. For Romania, this index is between 0.71 and 0.74, above the EU average [20]. According to the official prognosis, Romania is one of the EU member countries featuring a high potential risk of arable land abandonment due to factors related to biophysical land suitability, farm structure, agricultural viability, and population and regional specifics [11].

In this regard, the European Commission through the European Network for Rural Development Organization (ENRD) proposes policies and financing instruments for natural soil and water resources, as well as for traditional landscapes preservation and biological conservation.

Tourism as a saving factor for mountain communities is a myth from two decades ago [21,22], as we are now facing the effects of the replacement of the traditional, environmental-based economic system in mountain villages with the less adapted tourism-related activities [23]. Their effects are clearly visible in European mountain regions [24], including the later European Union membership states, featuring emerging rural tourism branch development [25].

Traditional mountain agriculture has followed a continuous negative trend since 1990, inversely correlated with the increasing tourism-based economy [26]. The replacement of traditional land use with other structures, based on rapid business development periods, meant cultural landscape loss and a decrease in sustainability [27], as well an increasing land degradation, as in Mediterranean mountain regions [28,29].

In the Carpathian region, traditional agriculture practices decreased after 1990, having a delay of about 20-30 years comparing to the Alpine region. Previous contributions highlight this process on the entire Carpathian ecoregion area [30–32], as well as on different case studies from Poland [33–35], Ukraine [36], and Slovakia [37].

In Romania, this process followed the same trend as in Central and Eastern Europe [31]. The agricultural lands from the mountainous area were not the subject of the centralized economy during the socialist period because of the specific environmental conditions [32]. Therefore, the agricultural land use change in the mountain areas was impacted mainly by socio-economic facts instead of political events [30].

The aim of our study is to quantify the arable land use changes between 1968 and 2018 in the Romanian Carpathians as a part of the mountain agricultural abandonment phenomenon that generates potential environmental conflicts.

## 2. Materials and Methods

Our analysis is applied within two representative case studies—Poiana Mărului, at 900–1000 m and Fundata, at 1300–1400 m altitude (Figure 1)—revealing a general feature for the entire Romanian Carpathians region. Both case studies are complementary from different points of view: altitude, topoclimate, tourist potential and valorization, entrepreneurship and business development, and regional branding. They are featured by autarky of human life, with arable land plots placed around farms, in severe bioclimatic and soil conditions, managed by the mountain farmers in a sustainable formula for the preservation of terrain productivity. The traditional crop rotation system [38], with a cycle of 7–10 years for the same crop type, put its imprint upon the landscape and can be considered to be a mark of a sustainable land use within the village agricultural land area.

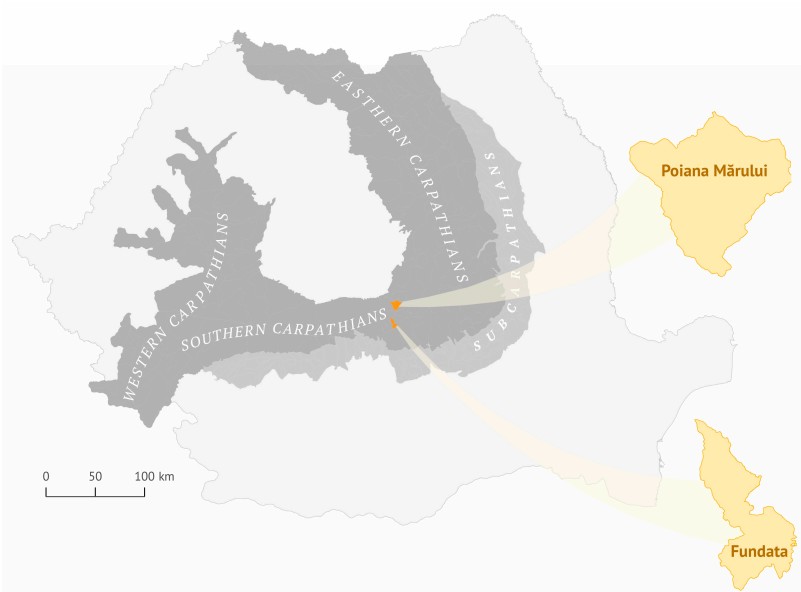

**Figure 1.** Case study areas and their geographical setting in Romania.

Our approach is based on multisource remote sensing image analysis for two reference periods combined with a spatial analysis of selected landscape metrics indices. In this context, we integrated a 2.0–3.0 m spatial resolution CORONA KH-4B DECLASS-1 (1968) satellite panchromatic dataset and Planet Scope satellite multispectral imagery from 2018, together with in-situ measurements obtained with the help of a geodetic GNSS (Global Navigation Satellite System) surveying (Table 1).

**Table 1.** Data types and data sources.

| Data Type | Name | Description | Data Source |
|---|---|---|---|
| Remote sensing images | CORONA KH-4B | 03.05.1968 Ca. 2.0 m, declassified data, panchromatic 02–03.05. 2018 | USGS, Declass 1 |
| | Planet Scope | 3.0 m, multispectral, visible+NIR | Planet Explorer |
| Field survey data | Arable land patch polygons – active, inactive | Leica Zeno 20 DGPS survey points, centimeter precision | Fieldwork September 2018, August 2019 |
| Statistics of population | Census data | Total population; active population in agriculture; employment; migration; | National Statistics Institute |
| Statistics of tourism | Census data | Accommodation capacity; tourist arrivals; | National Statistics Institute |
| Statistics of arable land and agriculture | Agricultural census data | Arable land (ha); Potato cultivated land (ha) | National Statistics Institute |

Reference images cover a temporal frame of 50 years, acquired at the beginning of the month of May, corresponding to the start of the agricultural season, when the spectral signature of arable plots is easily identified after the ploughing of soil.

A combined remote sensing and statistical approach is focused on mapping and evaluating the change of the arable land pattern.

The first image (1968) is a panchromatic, single-channel record from a declassified data archive from USGS, having a limited data amount, as the grayscale image at 8 bits depth is a restriction for semi-automatic supervised classification of arable land plots (patches). An interpretation process

based on grayscale level enhancement and visual texture analysis allowed for the production of all the polygons of arable land plots (Figure 2).

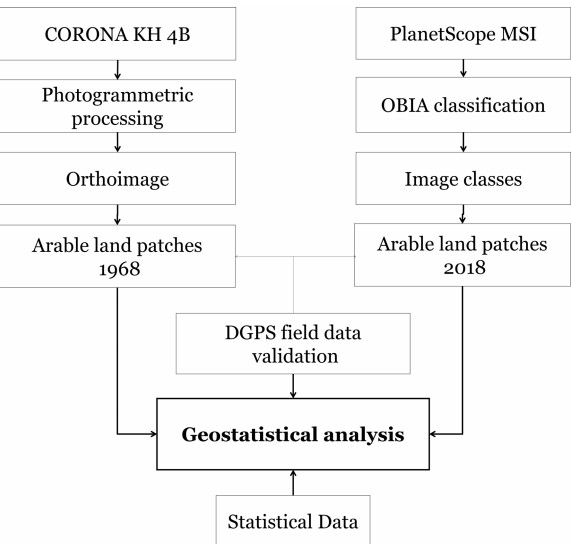

**Figure 2.** Workflow of the analysis.

The recent image from the Planet Scope archive (2018) is more consistent in information, as the multispectral channels cover the visible and the near infrared intervals at 3.0 m resolution. An object-based analysis (OBIA) is developed in order to generate the segmentation of the arable patches and their definition in terms of spectral signature differences in comparison with pastures, forests, and rock outcrops.

After performing the image calibration and orthorectification together with atmospheric and geometric corrections, two diachronic polygon themes (1968 and 2018) representing arable land plots are produced for each test area. These were processed in order to calculate the rate of occurrence of the arable land in square meters per hectare. A difference map of arable land for each study area is done together with the computation of the landscape metrics indices.

In-situ measurements were collected within two field campaigns (September 2018 and August 2019) using GNSS device, consisting of point datasets corresponding to active (cultivated) and abandoned plots to be restored on maps. These were used for land cover classification and for the validation of results.

Ancillary statistics focused on population, agriculture, and tourism were integrated in order to correlate the results of the current analysis.

## 3. Results

Two spatial datasets of the arable land occurrence (square meters per hectare) were obtained for both study sites in 1968 and 2018.

Figure 3a (1968) and Figure 3b (2018) show the arable land pattern for Poiana Mărului site. For 1968, high arable land densities over 2500 sq m/ha were typical for almost the entire scattered village area, counting 204 sample units, where traditional agriculture was still preserved, although the industrial development of the Brașov region also required a labor force from the rural communities [26]. For the second time (2018), advanced rural abandonment is evident, as the highest arable land density falls down to 1000–2000 sq m/ha in a highly dispersed pattern on less than 20 sample units (18).

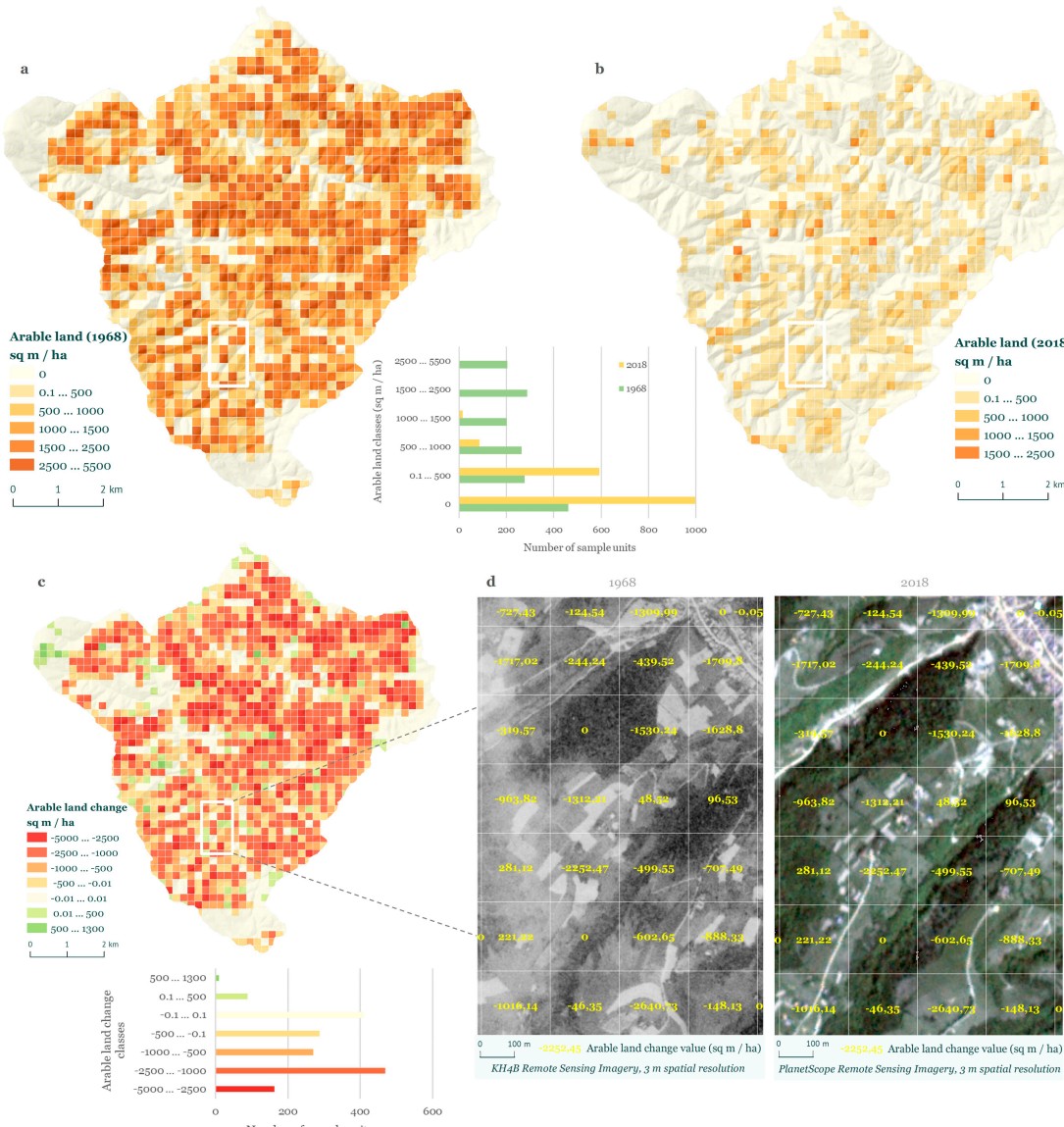

**Figure 3.** Arable land density in Poiana Mărului in 1968 (**a**) and 2018 (**b**). A difference map (**c**) and a sample of diachronic satellite image pair (**d**) confirm arable land use abandonment.

The general change trend between the reference times showed a drastic reduction of arable land surfaces. Arable land loss for the investigated 162 sample units are of more than 2500 sq m/ha, with the highest values of about 5000 sq m/ha (Figure 3c). A characteristic sample for this site is presented in Figure 3d, where arable plot patterns for each of the reference times can be seen in the same map together with the magnitude of changes.

Figure 4a (1968) and Figure 4b (2018) show the arable land pattern for the Fundata site. In 1968, the landscape preserved six classes of arable land density (similar to Poiana Mărului), featured in some points by values higher than 2000 sq m/ha. For 2018, only two classes were preserved at their lowest values, as there is no sample unit with more than 300 sq m of arable land per hectare. Figure 4c,d reveal a deep change of arable land use pattern, where most of traditional plots no longer preserve their function.

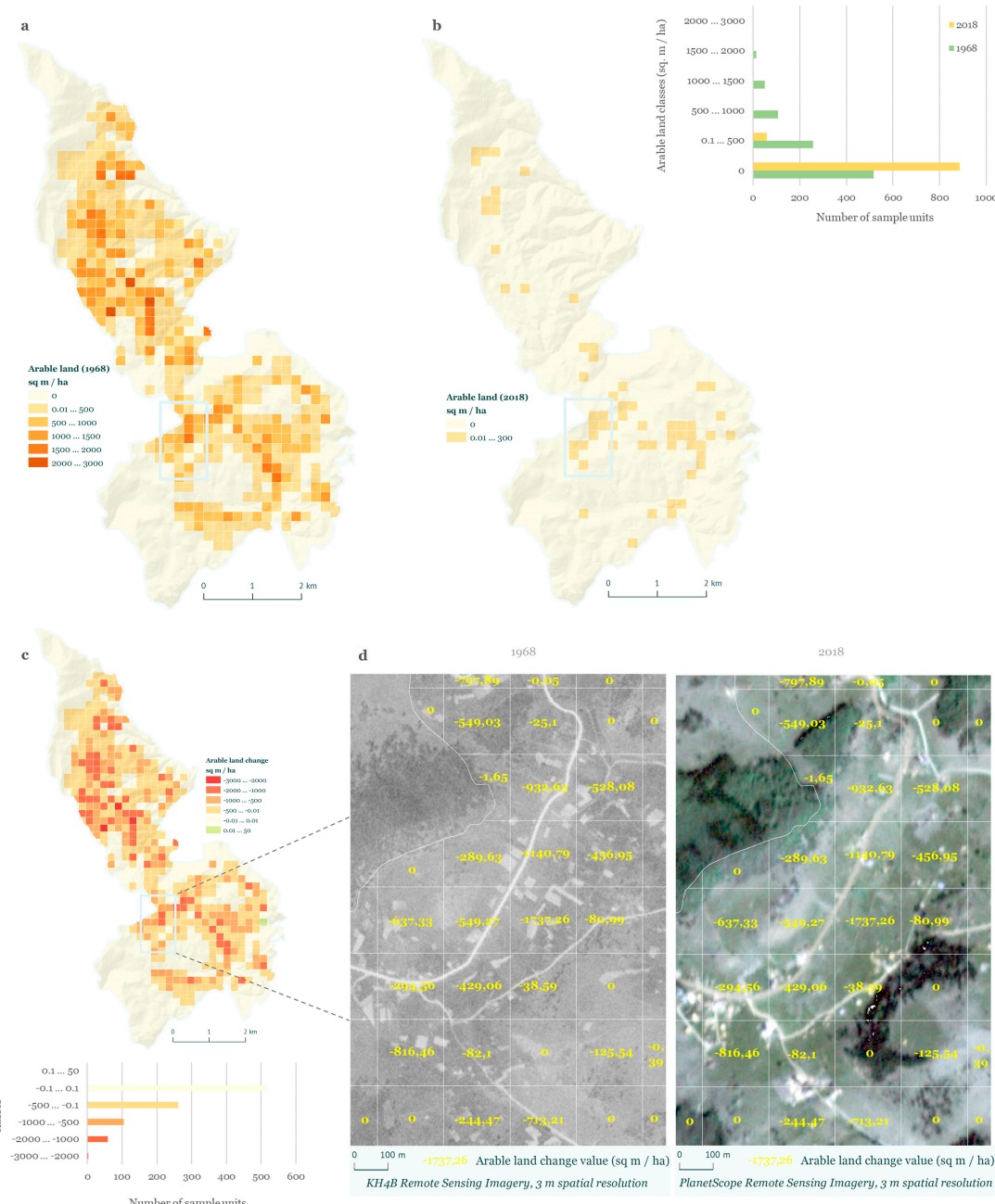

**Figure 4.** Arable land density in Fundata in 1968 (**a**) and 2018 (**b**). A difference map (**c**) and a sample of diachronic satellite image pair (**d**) confirm arable land use abandonment.

For a quantitative overview of arable land change, a set of various landscape metrics [39] is calculated. Table 2 shows, by comparison, the most relevant landscape metrics indices, together with the difference between them, confirming a general trend of landscape simplification, with the arable land extinction.

**Table 2.** Spatial metrics for arable land change analysis.

| Spatial Metrics | Units | Poiana Mărului | | | Fundata | | |
|---|---|---|---|---|---|---|---|
| | | 1968 | 2018 | 2018−968 | 1968 | 2018 | 2018−1968 |
| Number of arable patches | - | 1842 | 928 | −914 | 980 | 79 | −901 |
| Arable patch density | no/ 100 ha | 25.32 | 15.54 | −9.78 | 27 | 1.898 | −25.1016 |
| Arable patch area - *Mean* | ha | 0.445 | 0.08 | −0.3646 | 0.1 | 0.02 | −0.0795 |
| Arable patch area - *Area-weighted mean* | ha | 0.979 | 0.155 | −0.8235 | 0.217 | 0.039 | −0.1776 |
| Percentage of landscape occupied by the arable class | % | 11.27 | 1.24 | −10.031 | 2.69 | 0.038 | −2.652 |

## 4. Discussion

The comparative approach between Poiana Mărului and Fundata sites highlights two different stages of arable land abandonment process: an initial stage related to an emergence of rural tourism developments (Poiana Mărului) and an advanced stage strongly related to a total abandonment of arable patches around the remained scattered farms and tourist pensions and hotels (Fundata).

Intensive social transformations—demographic, economic, and political—leave their mark upon the traditional rural life [40,41]. Currently, depopulation and the traditional agriculture abandonment represent a global interest for researchers and policy makers at different levels [42].

Official statistics data referring to socio-economic indicators explain the arable land use change process. Total population (Figure 5a) illustrates a negative trend for both study sites. Poiana Mărului village had a smooth demographic decrease in comparison with the Fundata site, where demographic decrease is abrupt as an influence of higher accessibility along the transcarpathian road from Brașov to Pitești (DN 73/E 574).

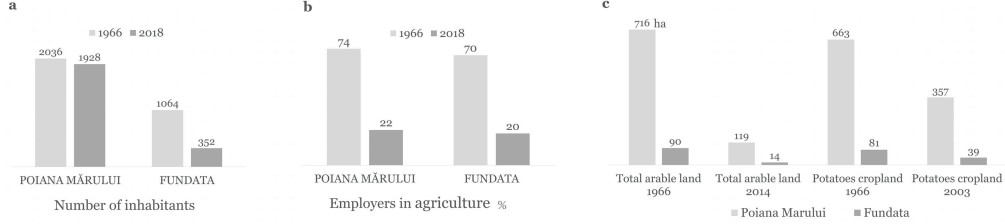

**Figure 5.** Statistical data featuring the socio-economic change in Poiana Mărului and Fundata for characteristic periods. (**a**) Population; (**b**) Employees in agriculture; (**c**) Arable cropland and potato cultivated land surface change in Poiana Mărului and Fundata administrative units. Statistical data source: National Institute of Statistics, Bucharest.

The percentage of employees in agricultural activities decreased by three times for both study areas (Figure 5b). An effect of demographic decrease and agricultural working force diminution is the extinction of arable land surface, together with the potato production, the main crop for the both sites. Statistics for potato crops show a decrease by 40–45%, beyond the conservative tradition preservation of the rural community (Figure 5c). Fundata and Poiana Mărului are by far examples of arable land abandonment by 7–9 times between 1966 and 2014.

The demographic and economic dynamics have a direct relationship with the abandonment of arable plots and the development of new tourism facilities, together with a new and modernized road network. Accommodation capacity statistics show a significant increase starting from 2007 for Fundata, while a similar situation took place in Poiana Mărului in 2017 (Figure 6a). Fundata illustrates the most dynamic situation, as the number of beds for tourism increased by 200% during the last 12 years.

From this point of view, Poiana Mărului village was a more conservative community, as the tourist accommodation capacity remained under 1000 beds.

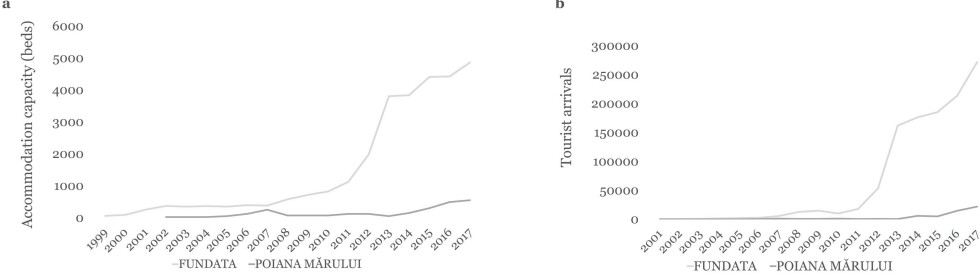

**Figure 6.** Statistical data showing the tourism features for Poiana Mărului and Fundata. (**a**) Accommodation capacity; (**b**) Tourist arrivals. Statistical data source: National Institute of Statistics, Bucharest.

Tourist arrivals (Figure 6b), as an indicator of the economic change of the mountain rural communities, increased by five times after 2007 for Fundata, where the development continued in an intensified rhythm after the economic crisis. Under this increase of tourism, local agriculture entered into an advanced stage of abandonment, as shown by the rare arable plots and the uniformization of the land use, including new houses and roads. The same situation but with lower magnitude took place in Poiana Mărului, with a constant increase of tourism after 2012. Statistical indicators—accommodation capacities and the number of tourist arrivals—outline a 10-year difference between the study sites.

Tourism accelerated development is illustrated by the increase of tourist arrivals between 2007 and 2017 by 50 times at Fundata and 20 times at Poiana Mărului, together with a local demographic decrease. These problems had a destabilizing effect upon the relationships within the local to regional spatial systemic structure population/inhabitants, traditional economy/activities, and the natural ecosystem (Figure 7). Tourist investments originate from outside the region, and these investors have no relationship with the areas and a less complete perception to the reality of these environments. In this context, the decrease to extinction of sustainable agricultural activities like in the case of Fundata is the source of disequilibrium generating environmental conflicts [43,44]. An increasing number of visitors and tourism activities bring traffic, noise pollution, and open new issues for the sustainability of the local communities [45]—water supply and consumption, energy supply, communications-related investments, and the need to search for waste management solutions, even in remote areas from the main roads. Deforestation, biodiversity loss, water pollution, soil erosion, and noise are real problems that have a rapid and a profound negative effect upon the sustainable equilibrium (Figure 8).

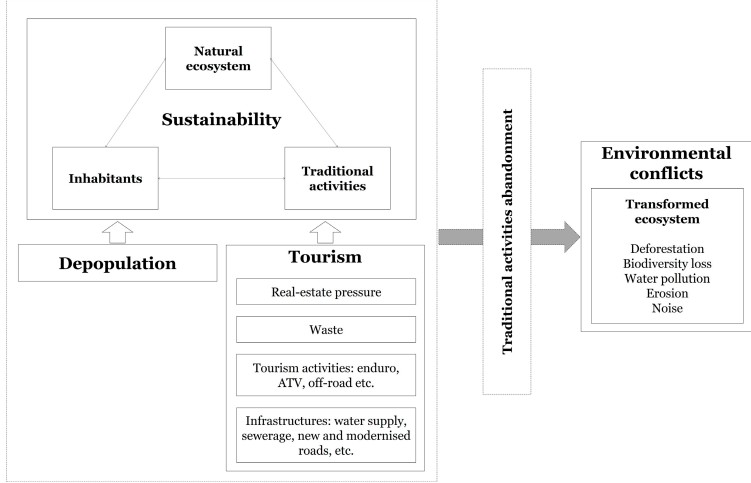

**Figure 7.** Environmental conflicts induced by the increasing tourism.

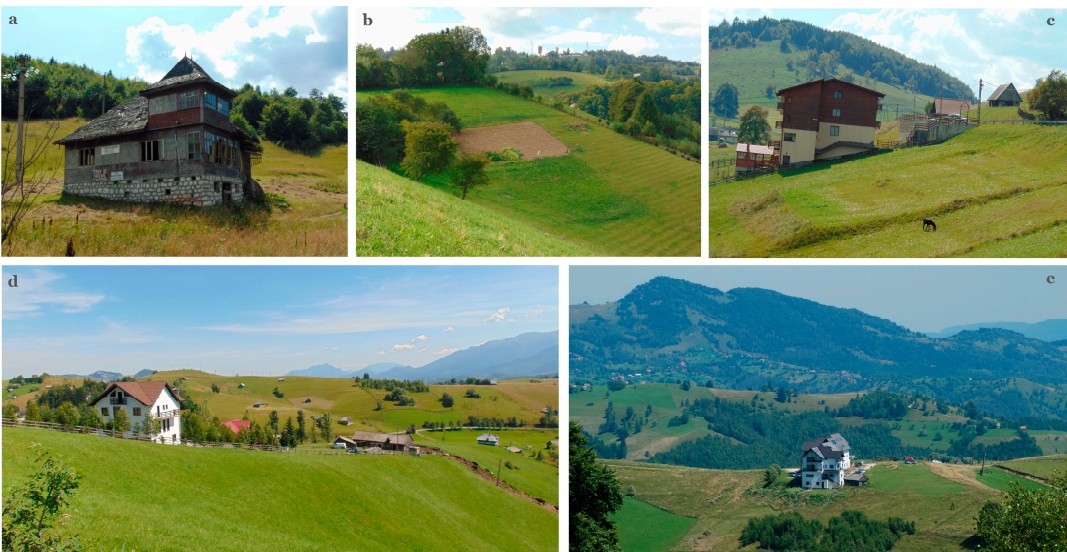

**Figure 8.** Different landscape and land use change issues in the study area: (**a**) Abandoned house in Fundata; (**b**) Decrease of arable plots in Poiana Mărului; (**c**) Recent buit-up area development and abandoned arable patches in Fundata; (**d**,**e**) Emerging hotels in Fundata.

## 5. Conclusions

Arable land abandonment in mountain areas is a worldwide problem affecting mountainous regions the world over. Our case studies, Poiana Mărului and Fundata, are characteristic examples of arable land abandonment in the Romanian Carpathians, in the context of a socio-economical system change after the EU integration of Romania in 2007.

Poiana Mărului and Fundata show different stages of arable land abandonment. Fundata reveals a profound change in the traditional economy in the context of a better developed infrastructure. Demographic processes like depopulation and population ageing, together with tourism developments, represent contributing factors to the decrease of the landscape diversity and to the increase of the mountain environment's dependency on the nearby urban economic system.

These socio-economical changes have direct effects on the functionality of the basic spatial system, where sustainable and traditional mountain agriculture is abandoned in favor of a service-based economy, including tourism-related activities. The traditional experience of land use is replaced step-by-step by the increasing volumes of investments, developed in a short time without a profound understanding of the environmental factors at local and regional scales. Environmental issues emerge from the abandonment of the traditional agricultural economy together with the development of new tourism facilities, hotels, access roads, and other infrastructures, creating new pressure factors upon the landscape.

**Author Contributions:** Conceptualization, I.S. and B.-A.M.; methodology, I.S., B.-A.M., M.V. and C.N.; software, M.V., C.N. and B.O.; validation, I.S., B.-A.M., M.V., C.N. and B.O.; formal analysis, I.S. and M.V.; investigation, I.S., M.V. and C.N.; resources, M.V., C.N. and B.O.; writing—original draft preparation, I.S., B.-A.M. and M.V.; writing—review and editing, I.S. and M.V.; visualization, M.V.; supervision, I.S.

**Funding:** This publication was funded by University of Bucharest, Faculty of Geography.

**Conflicts of Interest:** The authors declare no conflict of interest.

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
