# Peer review of "Mountain Arable Land Abandonment (1968–2018) in the Romanian Carpathians: Environmental Conflicts and Sustainability Issues"

_sustainability, doi:10.3390/su11236679_

Round 1

Reviewer 1 Report

Although the topic is interesting and worthy of being explored, the presentation is very poor, lacking research depth and very local in scope. In a nutshell, the article seems to be a report for the Romanian authorities, and not an article submitted to an international research journal. Even if analyzed as a local case study, the research seems to be immature, and does not reach even the declared goals of the study.

In more details, as the authors state, their research has only local value, both in terms of aims: "The aim of this study is to quantify the agricultural land use change occurred in low and medium altitude mountains, between 1968 and 2018, as an instrument for potential environmental conflicts assessments", and applicability: "This research could be a decision-making tool for local and regional levels sustainable development strategies". The conclusions, insufficiently developed, seem to be addressed to the Romanian authorities, and not to the broad research audience of Sustainability.

The declared goal (quantify the agricultural land use change as an instrument for potential environmental conflicts assessments) does not appear to be reached. The analysis does not seem to rely on the appropriate tools, or is not finished. Land use change is quantified, and the dynamic of some statistical indicators is presented (without being discussed), but the two are not related. I would have expected to find a synthetic table taking the discussion further, and identify potential conflicts by relating the two (land use change and dynamic of indicators).

The literature review is very poor and seems to be based on policies and not on the mainstream articles. Furthermore, critical concepts, announced by the title, such as "environmental conflicts" and appropriate sustainability issues, are not discussed in the introduction and discussions. The results are not validated by appropriate comparisons with similar case studies identified in the literature. As a result, the reference list is insufficiently developed.

In terms of writing, the article needs to be carefully revised in detail. First, the English needs to be assessed by a native speaker; in many instances, the order of words does not correspond to the proper phrasing in English, but is, most likely, a translation of words from a different language, without repositioning them properly. Second, the aims of research are declared in the middle of the introduction (rows 72-75) instead of concluding it. Third, the abstract is very poorly written and takes the form of a local research report. There are no details on the rationale beyond the research presented in the article. It simply states some results, without analyzing their significance through the lens of their contribution to the development of the field.

Author Response

The introductory section has been revised and partly rewritten in order to improve the research depth; Mainstream references have been included in order to update the literature list; The scope of the article has been rewritten, highlighting the importance of the issue on different scales: global, European, Carpathians ecoregion, Romanian Carpathians; The discussion section has been improved: statistical indicators have been interpreted according to arable land transformations; The potential environmental conflicts related to the arable land abandonment are illustrated within the scheme from Figure 7. We consider that a new table would double the information presented within the Figure 7 and the text, but if it is necessary, we could insert one; Critical concepts, like “environmental conflicts” and “sustainability issues” have been addressed in the introductory (Rows 48-50, 54-57, 66-70) and discussion (Rows 221-234) sections; The reference list has been updated with titles from similar approaches and case studies from Carpathian ecoregion; English language has been improved; The aim of the research is positioned to the end of the introductory section; The abstract has been rewritten according to the current configuration of the manuscript; Our contribution is the integration of high spatial resolution remote sensing data for the last 50 years, together with official statistics and field research and survey, in order to propose a more objective approach of the interface between arable land use abandonment in mountain areas and environmental conflicts; The title has been changed for a better understanding of the aim and scope of the paper. The new title is: “Mountain arable land abandonment (1968-2018) in Romanian Carpathians. Environmental conflicts and sustainability issues”;

Reviewer 2 Report

Major concerns:

Even the aim of the study (and the title) promises to quantify agricultural land use change, the results show only a simply message about the decrees of arable land in the study areas (which is already known from the statistical data). The added value of the remote sensing analyse is the spatial information about the arable land loss, however the spatial pattern of arable land abandonment is not further analysed or discussed. As mentioned above, we only know from the study that area of arable land has rapidly decreased, we don’t know what happened with it (how many of arable land was overgrowth by shrubs and trees, how much was transformed to grasslands and how the grasslands are managed, how much of the land was transformed to tourist infrastructure). If the aim of the paper is to quantify agricultural land use change, this needs to be investigated. If not, I recommend to change title and aim of the study focusing only on abandonment of arable land. But than, at least the spatial pattern of the abandonment needs to be analysed and discussed. The interesting part of the study is the use of CORONA images. I recommend to add “Corona” to keywords

Minor concerns:

L32 Do the traditional agriculture still plays a leading role in food security? L40 use of “abandonment intensification” is bit confusing Figure 2 is not needed, the workflow is sufficiently described in text The term “driving factor” is not appropriate. See Meyfroidt, P., 2016. Approaches and terminology for causal analysis in land systems science. Journal of Land Use Science 11, 501–522. https://doi.org/10.1080/1747423X.2015.1117530 L204 because the correlation was not calculated, do not use term “correlated”

Author Response

The title has been changed for a better understanding of the aim and scope of the paper. The new title is: “Mountain arable land abandonment (1968-2018) in Romanian Carpathians. Environmental conflicts and sustainability issues”. Our study focuses only on the arable land changes. Spatial pattern of arable land abandonment is illustrated for each study case within the Figures 3 and 4, also including charts. Corona and Planet Scope have been added to the keywords; Do the traditional agriculture still plays a leading role in food security? – We have been introduced new references addressing this issue (Rows 31-32); use of “abandonment intensification” is bit confusing – We have been rephrased; We consider that Figure 2 helps the readers for better understand the workflow of the analysis; We have been rewritten the text and we replaced “driving factor”; We have been replaced the term “correlated”;

Round 2

Reviewer 1 Report

The authors have carried out a substantial revision, addressing properly all comments. As a result, the article gained research depth and in its revised format addresses the broad international research audience of the journal. Therefore, I am fully recommending the revised submission for publication with "Sustainability".

Author Response

English language has been checked and minor corrections were brought to the manuscript.

Three references focused on farmland abandonment in mountain areas were added to the list.

Reviewer 2 Report

I have no more reccomendations.

Author Response

(The authors gave the same response as above.)
